# Preoperative Cancer Inflammation Prognostic Index as a Superior Predictor of Short- and Long-Term Outcomes in Patients with Stage I–III Colorectal Cancer after Curative Surgery

**DOI:** 10.3390/cancers14246232

**Published:** 2022-12-17

**Authors:** Jeng-Fu You, Yu-Jen Hsu, Yih-Jong Chern, Ching-Chung Cheng, Bor-Kang Jong, Chun-Kai Liao, Pao-Shiu Hsieh, Hung-Chih Hsu, Wen-Sy Tsai

**Affiliations:** 1Division of Colon and Rectal Surgery, Department of Surgery, Chang Gung Memorial Hospital at Linkou, College of Medicine, Chang Gung University, Taoyuan 33305, Taiwan; 2Division of Hematology-Oncology, Chang Gung Memorial Hospital at Linkou, College of Medicine, Chang Gung University, Taoyuan 33305, Taiwan

**Keywords:** colorectal cancer, cancer inflammation prognostic index, carcinoembryonic antigen, neutrophil-to-lymphocyte ratio, survival

## Abstract

**Simple Summary:**

Inflammatory reactions may lead to systemic inflammation and cancer growth. Some inflammatory indicators are effective predictors of colorectal cancer in ordinary clinical practice. The objective of this study is to evaluate the utility of a novel cancer-inflammation prognostic index (CIPI) marker derived from multiplying carcinoembryonic antigen by the neutrophil-to-lymphocyte ratio value obtained for non-metastatic colorectal cancer. Patients who underwent radical resection for stage I to stage III primary colorectal cancer between January 1995 and December 2018 were included in this study for further investigation. The group with a high CIPI had considerably lower relapse-free survival and overall survival rates, as well as a greater incidence of recurrence. Both univariate and multivariate analyses found that a high CIPI was an independent prognostic factor for survival analysis. This research is the first to demonstrate that CIPI is an independent factor that can be used to predict the outcome of colorectal cancer.

**Abstract:**

Inflammatory reactions play a crucial role in cancer progression and may contribute to systemic inflammation. In routine clinical practice, some inflammatory biomarkers can be utilized as valuable predictors for colorectal cancer (CRC). This study aims to determine the usefulness of a novel cancer-inflammation prognostic index (CIPI) marker derived from calculating carcinoembryonic antigen (CEA) multiplied by the neutrophil-to-lymphocyte ratio (NLR) values established for non-metastatic CRCs. Between January 1995 and December 2018, 12,092 patients were diagnosed with stage I to III primary CRC and had radical resection—they were all included in this study for further investigation. There were 5996 (49.6%) patients in the low-CIPI group and 6096 (50.4%) patients in the high-CIPI group according to the cutoff value of 8. For long-term outcomes, the high-CIPI group had a significantly higher incidence of recurrence (30.6% vs. 16.0%, *p* < 0.001) and worse relapse-free survival (RFS) and overall survival (OS) rates (*p* < 0.001). High CIPI was an independent prognostic factor for RFS and OS in univariate and multivariate analyses. This research is the first to document the independent significance of CIPI as a prognostic factor for CRC. To ensure that it works, this CIPI needs to be tested on more CRC prediction models.

## 1. Introduction

The tumor microenvironment is complex, containing cells with various properties; inflammatory cells are considered, generally, critical to the tumor microenvironment [1,2]. Inflammatory responses play an essential role in cancer evolution and progression and are possibly involved in systemic inflammation [3,4]. There is growing evidence that inflammatory markers are closely related to poor prognosis in colorectal cancer (CRC) [5,6,7,8]. One well-known marker of systemic inflammation is serum C-reactive protein (CRP) level [9,10]. Several studies have elucidated that high-CRP levels are associated with poorer survival in CRC patients and could be used as a prognostic biomarker [11,12,13,14,15]. In addition, combinations of parameters have been widely utilized to predict CRC outcomes, including neutrophil-to-lymphocyte ratio (NLR), platelet-to-lymphocyte ratio (PLR), lymphocyte-to-monocyte ratio (LMR), Glasgow prognostic score (GPS), prognostic nutritional index (PNI), and geriatric nutritional risk index (GNRI). These biomarkers can be used as valuable predictors in daily clinical work [16].

Carcinoembryonic antigen (CEA) is a glycoprotein and is currently the most widely implemented tumor marker in CRC patients. The primary role of this tumor marker is to detect recurrence as early as possible during follow-up visits after curative surgery for CRC [17,18,19]. The sensitivity of serum values of CEA at diagnosis is usually low because serum CEA levels can be within the normal range in patients with CRC. Several studies have analyzed the association between high preoperative serum levels of CEA and poorer prognosis [20,21,22]. Evidence suggests that patients with elevated serum levels of preoperative CEA may have a higher recurrence rate than patients with normal levels prior to the operation [23,24,25].

A new prognostic index, defined as the cancer-inflammation prognostic index (CIPI), based on CEA and NLR, was recently introduced as a promising novel prognostic marker for metastatic CRC patients treated with regorafenib [26,27]. To our knowledge, CIPI has never been utilized as a prognostic marker in non-metastatic CRC after curative resection. In theory, predicting prognosis after radical surgery for CRC involves evaluation of the interaction between patient- and tumor-related factors. Thus, it is reasonable to suggest the use of CIPI as a simultaneous surrogate for tumor- and immune-related markers. Therefore, in the present study, we investigate the prognostic significance of CIPI in patients with non-metastatic CRC after curative surgery.

This study was designed to explore the value of a novel marker, CIPI, by calculating CEA multiplied by NLR values developed for non-metastatic CRCs. To our knowledge, no previous analysis has assessed the significance of CIPI in non-metastatic CRC after curative resection. Therefore, we examine the prognostic value of CIPI to indicate recurrence risk and survival outcomes in patients with stage I–III CRC after radical resection in a real-world dataset.

## 2. Materials and Methods

### 2.1. Data Collection and Study Design

This retrospective study examined the records of a database from the Colorectal Section Tumor Registry of Chang Gung Memorial Hospital. The institutional review board approved this study (IRB No. 201601428B0).

The inclusion criteria were as follows: histologically confirmed colorectal adenocarcinoma and complete clinical, laboratory, imaging, and follow-up data. The exclusion criteria included: emergency operation, non-curative resection, and missing CEA or NLR data.

Database variables included age, sex, body mass index (BMI), underlying disease, preoperative routine serological test, and tumor-related variables. We collected laboratory data within one week prior to patients undergoing surgery. According to the 8th edition of Cancer Staging, all patients in this study were staged. 

### 2.2. Cancer-Inflammation Prognostic Index (CIPI)

CIPI is calculated as CEA (mg/L) × NLR, where NLR denotes the neutrophil-to-lymphocyte counts ratio. The receiver operating characteristic (ROC) curve was used to assess the ability of CIPI to classify disease status. The optimal cutoff value of CIPI for tumor recurrence was determined by ROC curve analysis according to the Youden index.

### 2.3. Measurement Outcomes

Measurement outcomes included short-term postoperative complications and long-term survival. Postoperative complications were defined as complications that occurred within 30 days of surgery. Postoperative mortality was defined as death that occurred within 30 days of surgery. Long-term outcomes were evaluated using relapse-free survival (RFS) and overall survival (OS). The first recurrence was defined as the first date when local recurrence or distant metastases were confirmed by histology of biopsy specimens, additional surgery, or radiological studies. RFS was defined as the interval between cancer resection and the first recurrence, death, or the final follow-up date. OS was defined as the interval between cancer resection and death or the date of the last follow-up.

### 2.4. Statistics

All analyses were conducted using SPSS Statistics, version 24.0 (IBM Corp, Armonk, NY, USA). Clinicopathological characteristics with categorical variables were presented as frequencies and proportions and were compared using the chi-squared test. Continuous variables were expressed as means and standard deviations and were analyzed using the Student *t*-test. RFS, OS, and time-to-event probabilities were determined and plotted using the Kaplan–Meier method. Differences were estimated using the log-rank test. Univariate and multivariate analyses of independent factors for RFS and OS were performed using the Cox proportional hazards regression analysis. We incorporated all variables with *p* < 0.05 from the univariate regression analysis into a multivariate Cox proportional hazards regression model to investigate the explanatory variables further. A *p*-value < 0.05 was considered statistically significant.

## 3. Results

Between January 1995 and December 2018, 13,954 patients with stage I to III primary CRC who had received radical resection were included in the study. Of these 13,954 patients, 104, 382, and 1376 were excluded because they had a non-curative resection, missing CEA data, and missing NLR data, respectively. The remaining 12,092 patients were included.

### 3.1. The CIPI Cutoff Level

The ROC curve that demarcates the predictivity of CIPI and five-year RFS is illustrated for the whole cohort of patients. As shown in Figure 1a, the area under the curve (AUC) was 0.658 (95% CI: 0.648–0.669; *p* = 0.001). Estimation using the Youden index showed that the recommended cutoff value of preoperative CIPI for evaluating five-year RFS was 8.351, with a sensitivity of 0.659 and a specificity of 0.585. To simplify grouping and application, we determined a recommended cutoff value of 8.0 for CIPI, with a sensitivity of 0.67 and a specificity of 0.569 for further study.

The CIPI value of all patients ranged from 0.3 to 10,909.4, with a mean value of 43.2 ± 224.5 and a median value of 8.1. The mean and 5% trimmed values of CIPI were 75.4 and 32.4, respectively, for patients with recurrence or mortality within five years; these values were 29.1 and 12.5 for those without recurrence or mortality within five years. The median CIPI value for patients with recurrence or mortality within five years was 13.7; this value was 6.5 for those without recurrence or mortality within five years (Figure 1b).

### 3.2. CIPI and Clinicopathological Features

There were 5996 (49.6%) patients in the low-CIPI group and 6096 (50.4%) patients in the high-CIPI group. Table 1 shows the relationships between CIPI and clinicopathological characteristics.

For the baseline clinical data, patients with high-CIPI were usually older (64.94 ± 13.12 vs. 62.10 ± 12.67 years, *p* < 0.001), more likely male (58.7% vs. 54.5%, *p* < 0.001), and had a lower BMI (<25 kg/m^2^: 67.4% vs. 59.3%, *p* < 0.001). For underlying medical illnesses, the high-CIPI group had a higher proportion of diabetes mellitus (18.9% vs. 14.1%, *p* < 0.001) and chronic renal disease (9.8% vs. 4.8%, *p* < 0.001) than the low-CIPI group. However, patients with a history of hepatitis had lower CIPI levels (4.3% vs. 5.4%, *p* = 0.006).

Regarding tumor-related characteristics, there was no significant difference between the high- and low-CIPI groups in terms of tumor location. However, the high-CIPI group had a more advanced tumor stage (*p* < 0.001), including T stage (*p* < 0.001), N stage (*p* < 0.001), histologic type (*p* < 0.001) and histologic differentiation (*p* < 0.001). CIPI was also associated with obstruction (*p* < 0.001), perforation (*p* < 0.001), and preoperative laboratory values, including hemoglobin (*p* < 0.001) and albumin (*p* < 0.001) levels.

### 3.3. CIPI and Short- and Long-Term Outcomes

Details of the association between CIPI and short- and long-term outcomes are presented in Table 2. For short-term outcomes, the preoperative high-CIPI group had a higher postoperative 30-day morbidity and mortality than the low-CIPI group, both *p* < 0.001.

For long-term outcomes, the high-CIPI group had a significantly higher incidence of recurrence (30.6% vs. 16.0%, *p* < 0.001) and worse RFS and OS (*p* < 0.001). The three-year, five-year, and 10-year RFS rates were 81.5%, 77.3%, and 68.2%, respectively, in the low-CIPI group and 61.8%, 56.2%, and 45.5%, respectively, in the high-CIPI group (*p* < 0.001). Similarly, OS followed the same trend of worse survival rates in the high-CIPI group compared to the low-CIPI group considering three-year, five-year, and 10-year survival.

### 3.4. CIPI and Survival Analyses According to Tumor Stage

In all patients with TNM stage I to III CRC, the high-CIPI group had lower RFS and OS rates than the low-CIPI group (*p* < 0.001), as displayed in Figure 2. These patients were divided into three subgroups according to their TNM stage: stage I (*n* = 2527), stage II (*n* = 4471), and stage III (*n* = 5094).

In TNM stage I patients, the estimated 10-year RFS rates of the low-CIPI and high-CIPI groups were 79.9% and 56.6%, respectively (*p* < 0.001). In the TNM stage II group, the estimated 10-year RFS rates of the low-CIPI and high-CIPI groups were 68.1% and 51.8%, respectively (*p* < 0.001). In the stage III group, the estimated 10-year RFS rates of the low-CIPI and high-CIPI groups were 58.3% and 37.6%, respectively (*p* < 0.001). The results of the OS analyses of each TNM stage are also presented in Figure 2. Patients in the high-CIPI group had lower OS at each tumor stage than those in the low-CIPI group (*p* < 0.001).

### 3.5. Univariate and Multivariate Analysis of RFS and OS

Detailed univariate and multivariate analysis results for RFS are listed in Table 3.

For the univariate analysis, age, gender, BMI, hypertension, cardiovascular disease, diabetes mellitus, hepatitis, chronic renal disease, tumor location, TNM stage, pT status, pN status, histologic type, histologic grade, obstruction, perforation, hemoglobin level, albumin level, and CIPI were significantly associated with RFS.

The multivariate analysis showed that age ≥ 65 years (HR: 1.59, *p* < 0.001), male sex (HR: 1.252, *p* < 0.001), hypertension (HR: 1.131, *p* < 0.001), cardiovascular disease (HR: 1.26, *p <* 0.001), diabetes mellitus (HR: 1.147, *p* = 0.001), chronic renal disease (HR: 1.447, *p* < 0.001), rectum (HR: 1.493, *p* < 0.001), pT2 (HR: 1.45, *p* < 0.001), pT3 (HR: 1.841, *p* < 0.001), pT4 (HR: 2.542, *p* < 0.001), pN1 (HR: 1.46, *p* < 0.001), pN2 (HR: 2.421, *p* < 0.001), signet ring cell adenocarcinoma (HR: 1.452, *p* = 0.016), obstruction (HR: 1.276, *p* < 0.001), hemoglobin ≥ 10 g/dL (HR: 0.873, *p* < 0.001), albumin ≥ 3.5 g/dL (HR: 0.658, *p* < 0.001), and CIPI ≥ 8 (HR: 1.478, *p* < 0.001) were independent prognostic factors for RFS.

The specific univariate and multivariate analysis results of OS are presented in Table 4, showing a considerable similarity with RFS.

### 3.6. Subgroup Analyses of Survival Based on CIPI

Figure 3 illustrates the subgroup analysis to investigate whether the effect of clinicopathological variables on survival changed according to CIPI level. Patients in the high-CIPI group invariably had poorer RFS and OS than those in the low-CIPI group, except those with signet ring cell adenocarcinoma (RFS: HR 1.749, CI 0.948–3.224, *p* = 0.074; OS: HR 1.714, CI 0.928–3.166, *p* = 0.085). When stratified by pathological stage, HR values of the high-CIPI group for RFS and OS were higher in stage I than in stages II and III compared to the low-CIPI group.

## 4. Discussion

To the best of our knowledge, this is the first study to demonstrate an association between CIPI and the prognosis of non-metastatic CRC following curative resection. Our data suggest that the CIPI is a promising new prediction tool for CRC patients following curative resection based on CEA and NLR. The predictive value of CIPI is significant across all tumor stages and outperformed other serum marker indicators in predicting tumor recurrence and survival outcomes. In particular, CIPI is more significant for predicting the recurrence and survival outcome, especially in stage I colorectal cancer after surgery. Theoretically, determining the prognosis of CRC after radical surgery necessitates the examination of factors related to the patient and the tumor. CIPI serves as a kind of approximation for tumor- and immune-related indicators.

CEA is a member of the immunoglobulin supergene family expressed in normal mucosa cells with the biological function of cell recognition or adhesion mechanisms [28]. An animal study of CEA knockdown in mice reported a 50% decrease in metastatic ability, which indicated its influence on metastatic progression [29]. The elevation of serum CEA itself may also cause pro-inflammatory cytokines and induce adhesion molecules (E-selectin, ICAM-I, and VCAM), thus leading to the metastasis of CRC [30]. CEA is a helpful tumor marker in colorectal cancer. Its increase has been linked to a worse prognosis for CRC [31,32,33]. However, the optimal cutoff level of CEA elevation applied to predict recurrence is inconsistent under different conditions, and even some benign conditions influence its elevation. Generally, a cutoff level of 5 ng/mL is used to define CEA elevation. However, a preoperative CEA of 10 ng/mL has been advanced as the optimal level for assessing recurrence risk in CRC patients with stage II and III, but not stage I, according to a study of 638 stage I–III CRC patients after curative surgery [34]. Other studies suggest that optimal levels of high risk for recurrence include 6 ng/mL in stage IIA CRC patients who underwent colectomy surgery [35] and 2.5 ng/mL in stage I–III rectal cancer patients who underwent both chemoradiotherapy and surgery [36].

CEA elevation was also found in non-malignant conditions such as cirrhosis, ulcerative colitis, chronic renal failure, hypothyroidism, pancreatitis, and chronic lung disease [37,38,39,40]. Other metabolic and lifestyle factors, such as age, fasting glucose, cigarette smoking, and alcohol consumption, were also found to be associated with CEA elevation [41]. Although the level of CEA elevation engendered by benign factors has rarely reached over 10 ng/mL, a level of below 10, especially 5–10 ng/mL, has some limitations for predicting recurrence.

Inflammation and infection are other biological markers with a negative impact on prognosis. Chronic inflammation increases the risk of developing cancer and is inextricably linked to the immune system [42]. Inflammation and cancer use similar mechanisms; the microenvironment contributes to proliferation and survival of malignant cells, angiogenesis, metastasis, inhibition of adaptive immunity, response to hormones, and chemotherapeutic agents [43]. Neutrophils, lymphocytes, monocytes, and platelets are cellular components of systemic inflammation. Previous research has found that lymphocytes have a significant role in cancer immune surveillance, inhibiting the proliferation of tumor cells, restraining metastasis through cytokine production, and inducing cytotoxic cell death [44]. However, neutrophils, the first recruited effectors of the acute inflammatory response, may inhibit the immune system by suppressing the cytolytic activity of immune cells such as lymphocytes, activated T cells, and natural killer cells. They may also produce proangiogenic factors such as vascular endothelial growth factors, proteases, and chemokines and enhance the adhesion of circulating tumor cells in distant sites [45].

Two different types of scores have been proposed to track the systemic inflammatory response: those based on protein measurement and those based on inflammatory cell counting. Patients with CRC can have their condition predicted by counting these differentiated cells individually or collectively [46,47,48,49,50]. Studies evaluating the relationship between NLR and various cancers have shown that it was a strong prognostic factor [51,52]. In colorectal cancer, elevated NLR was associated with worse outcomes, and it has also been extensively discussed with worse prognosis in multiple treatment subgroups [53,54,55]. In our institution, the predictive value of NLR for CRC was published. According to our data, a high NLR was linked to a higher risk of RFS in colon cancer, but a multivariate analysis showed that it was less crucial in rectal cancer [56,57]. The subgroup analysis of this study revealed that patients in the high-CIPI group invariably had worse RFS and OS than those in the low-CIPI group, except for individuals with signet ring cell adenocarcinoma. Our data indicate that a high CIPI exhibited poor outcomes in both colon and rectal cancer, whereas a high NLR was not associated with poor outcomes in rectal cancer. From our data, CEA plus the inflammation index was found to predict tumor recurrence better than CEA or the inflammation index alone. Combining CEA and NLR may result in more valuable and extensive therapeutic applications for various colorectal cancer prognosis predictions.

This study has several limitations. First, the retrospective design of this study may increase the risk of bias. Second, CIPI values may be missing; for example, in the case of emergency surgery, there may be no CEA data prior to surgery. Third, presence of underlying illness may significantly impact the CIPI value, especially when the body is in an inflammatory state. Fourth, CIPI values are not a fixed, absolute value—in other words, even if the blood test is repeated at a comparable time, the CIPI values may be close but not the same, which will inevitably cause bias. Therefore, it might be reasonable to assume that using a trajectory trend of CIPI values would be more practical than a single-point value.

## 5. Conclusions

In conclusion, the results of the present study suggest that CIPI can function as an effective and easy-to-use clinical tool for patients with non-metastatic CRC after curative resection. This investigation is the first report to reveal that CIPI is an independent significant prognostic factor for resected CRC. Further studies applying CIPI to various prognostic models of CRC are required to ensure its usefulness.

## Figures and Tables

**Figure 1 cancers-14-06232-f001:**
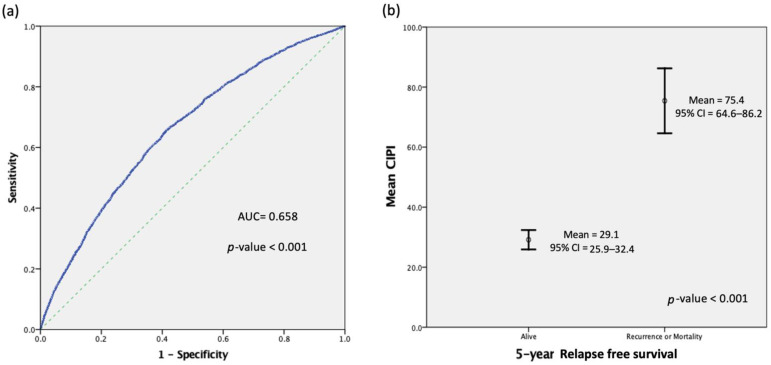
(**a**) Receiver operating characteristic (ROC) curve analysis of the cancer-inflammation prognostic index (CIPI) for five-year relapse-free survival (RFS) in stage I–III colorectal cancer (CRC). (**b**) Mean CIPI of patients alive and with recurrence or mortality during five-year follow-up. AUC = area under the curve; CI = confidence interval.

**Figure 2 cancers-14-06232-f002:**
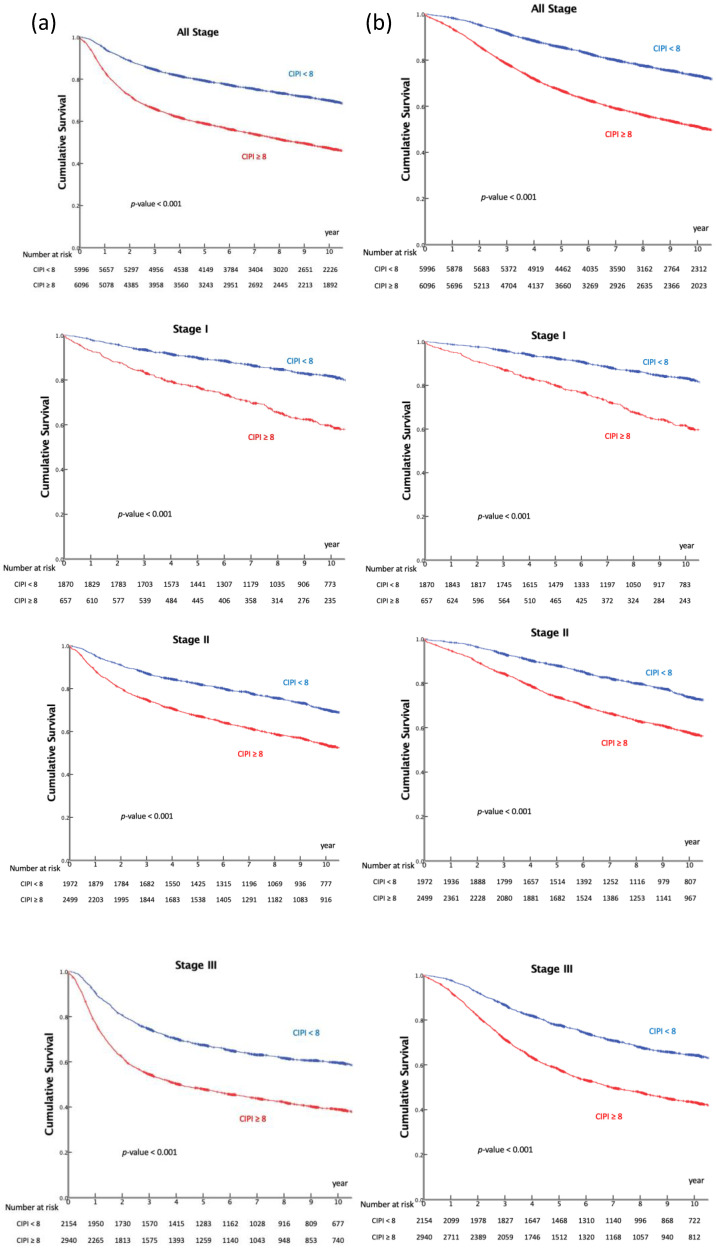
Kaplan–Meier survival analysis for relapse-free survival (RFS) (**a**) and overall survival (OS) (**b**) according to cancer-inflammation prognostic index (CIPI).

**Figure 3 cancers-14-06232-f003:**
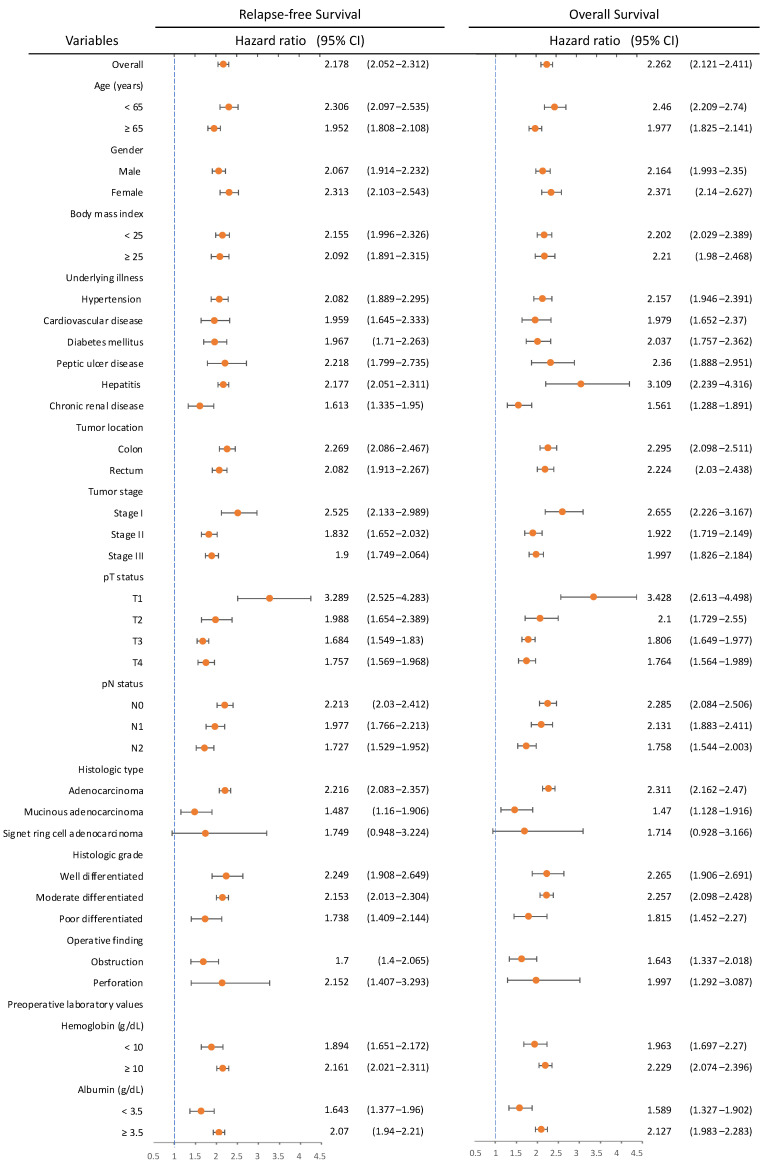
Subgroup analyses of relapse-free survival and overall survival based on cancer-inflammation prognostic index (CIPI) level.

**Table 1 cancers-14-06232-t001:** Demographic characteristics of patients based on the CIPI.

Variables	CIPI (*n* = 12,092)		*p* Values
	Low < 8 (*n* = 5996)	High ≥ 8 (*n* = 6096)	
Age, mean ± SD	62.10 ± 12.67	64.94 ± 13.12	<0.001
Age (years), *n* (%)			
<65	3331 (55.6)	2798 (45.9)	<0.001
≥65	2665 (44.4)	3298 (54.1)	
Gender, *n* (%)			
Male	3267 (54.5)	3581 (58.7)	<0.001
Female	2729 (45.5)	2515 (41.3)	
Body mass index, kg/m^2^, *n* (%)			
<25	3509 (59.3)	3970 (67.4)	<0.001
≥25	2407 (40.7)	1916 (32.6)	
missing	80	210	
Underlying illness, *n* (%)			
Hypertension	1978 (33.0)	2045 (33.5)	0.493
Cardiovascular disease	511 (8.5)	550 (9.0)	0.376
Diabetes mellitus	843 (14.1)	1153 (18.9)	<0.001
Peptic ulcer disease	447 (7.5)	458 (7.5)	0.993
Hepatitis	326 (5.4)	263 (4.3)	0.006
Chronic renal disease	282 (4.8)	586 (9.8)	<0.001
Tumor location, *n* (%)			
Colon	3440 (57.4)	3411 (56.0)	0.119
Rectum	2556 (42.6)	2685 (44.0)	
Tumor stage, *n* (%)			
Stage I	1870 (31.2)	657 (10.8)	<0.001
Stage II	1972 (32.9)	2499 (41.0)	
Stage III	2154 (35.9)	2940 (48.2)	
pT status, *n* (%)			
T1	1129 (18.8)	276 (4.5)	<0.001
T2	1094 (18.2)	518 (8.5)	
T3	2842 (47.4)	3097 (50.8)	
T4	931 (15.5)	22.5 (36.2)	
pN status, *n* (%)			
N0	3842 (64.1)	3153 (51.7)	<0.001
N1	1394 (23.2)	1738 (28.5)	
N2	760 (12.7)	1205 (19.8)	
Histologic type, *n* (%)			
Adenocarcinoma	5733 (95.6)	5611 (92.0)	<0.001
Mucinous adenocarcinoma	238 (4.0)	432 (7.1)	
Signet ring cell adenocarcinoma	25 (0.4)	53 (0.9)	
Histologic grade, *n* (%)			
Well differentiated	1063 (17.7)	767 (12.6)	<0.001
Moderate differentiated	4515 (75.3)	4814 (79.0)	
Poor differentiated	362 (6.0)	498 (8.2)	
Unclassified	56 (0.9)	17 (0.3)	
Operative finding			
Obstruction	312 (5.2)	814 (13.4)	<0.001
Perforation	70 (1.2)	314 (5.2)	<0.001
Preoperative laboratory values, *n* (%)			
Hemoglobin (g/dL)			
<10	743 (12.4)	1429 (23.5)	<0.001
≥10	5247 (87.6)	4663 (76.5)	
missing	6	4	
Albumin (g/dL)			
<3.5	303 (5.1)	1065 (17.9)	<0.001
≥3.5	5630 (94.9)	4891 (82.1)	
missing	63	140	

**Table 2 cancers-14-06232-t002:** Association of CIPI with short- and long-term outcome.

Variables	CIPI (*n* = 12,092)		*p* Values
	Low < 8 (*n* = 5996)	High ≥ 8 (*n* = 6096)	
Short-term postoperative outcome, *n* (%)			
30-day morbidity	660 (11.0)	1032 (16.9)	<0.001
30-day mortality	14 (0.2)	73 (1.2)	<0.001
Long-term outcome, *n* (survival%)			
Recurrence	961 (16.0)	1865 (30.6)	<0.001
three-year RFS	4956 (81.5)	3958 (61.8)	<0.001
three-year OS	5372 (88.4)	4704 (72.0)	<0.001
five-year RFS	4149 (77.3)	3243 (56.2)	<0.001
five-year OS	4462 (82.9)	3660 (62.7)	<0.001
10-year RFS	2226 (68.2)	1892 (45.5)	<0.001
10-year OS	2312 (71.5)	2023 (49.1)	<0.001

CIPI = cancer-inflammation prognostic index; RFS = relapse-free survival; OS = overall survival.

**Table 3 cancers-14-06232-t003:** Univariate and multivariate Cox proportional hazards regression analyses of prognostic factors for relapse-free survival.

Characteristics	Univariate			Multivariate		
	HR	95% CI	*p* Values	HR	95% CI	*p* Values
Age						
<65	1			1		
≥65	1.814	1.711–1.923	<0.001	1.59	1.49–1.696	<0.001
Gender						
Female	1			1		
Male	1.251	1.180–1.326	<0.001	1.252	1.176–1.333	<0.001
Body mass index						
<25	1			1		
≥25	0.877	0.825–0.933	<0.001	0.937	0.877–1.001	0.053
Underlying illness						
Hypertension	1.229	1.163–1.299	<0.001	1.131	1.060–1.206	<0.001
Cardiovascular	1.381	1.282–1.488	<0.001	1.26	1.150–1.381	<0.001
Diabetes mellitus	1.33	1.238–1.428	<0.001	1.147	1.060–1.242	0.001
Peptic ulcer	1.053	0.958–1.158	0.282			
Hepatitis	0.802	0.697–0.924	0.002	0.945	0.821–1.087	0.427
Renal disease	2.006	1.835–2.193	<0.001	1.447	1.308–1.599	<0.001
Tumor location						
Colon	1			1		
Rectum	1.31	1.237–1.386	<0.001	1.493	1.401–1.591	<0.001
Tumor stage						
Stage I	1					
Stage II	1.821	1.653–2.006	<0.001			
Stage III	3.008	2.743–3.300	<0.001			
pT status						
T1	1			1		
T2	1.818	1.550–2.132	<0.001	1.45	1.221–1.721	<0.001
T3	2.823	2.463–3.236	<0.001	1.841	1.577–2.148	<0.001
T4	4.418	3.846–5.076	<0.001	2.542	2.165–2.986	<0.001
pN status						
N0	1			1		
N1	1.602	1.498–1.714	<0.001	1.46	1.357–1.572	<0.001
N2	2.777	2.587–2.980	<0.001	2.421	2.236–2.621	<0.001
Histologic type						
Adenocarcinoma	1			1		
Mucinous	1.208	1.075–1.357	0.001	0.978	0.856–1.117	0.740
Signet ring cell	2.621	2.004–3.428	<0.001	1.452	1.071–1.968	0.016
Histologic grade						
Well	1			1		
Moderate	1.37	1.256–1.494	<0.001	0.966	0.875–1.066	0.488
Poor	1.743	1.535–1.980	<0.001	1.014	0.876–1.175	0.848
Operative finding						
Obstruction	1.778	1.632–1.936	<0.001	1.276	1.159–1.404	<0.001
Perforation	1.769	1.540–2.033	<0.001	1.145	0.980–1.337	0.087
Preoperative values						
Hemoglobin						
<10	1			1		
≥10	0.657	0.614–0.703	<0.001	0.873	0.807–0.945	<0.001
Albumin						
<3.5	1			1		
≥3.5	0.471	0.438–0.508	<0.001	0.658	0.601–0.720	<0.001
CIPI						
<8	1			1		
≥8	2.178	2.052–2.312	<0.001	1.478	1.383–1.579	<0.001

CIPI = cancer-inflammation prognostic index.

**Table 4 cancers-14-06232-t004:** Univariate and multivariate Cox proportional hazards regression analyses of prognostic factors for overall survival.

Characteristics	Univariate			Multivariate		
	HR	95% CI	*p* Values	HR	95% CI	*p* Values
Age						
<65	1			1		
≥65	2.253	2.114–2.401	<0.001	1.958	1.825–2.097	<0.001
Gender						
Female	1			1		
Male	1.282	1.205–1.364	<0.001	1.286	1.204–1.375	<0.001
Body mass index						
<25	1			1		
≥25	0.834	0.781–0.890	<0.001	0.889	0.829–0.953	0.001
Underlying illness						
Hypertension	1.353	1.271–1.439	<0.001	1.189	1.107–1.277	<0.001
Cardiovascular	1.648	1.502–1.807	<0.001	1.345	1.219–1.485	<0.001
Diabetes mellitus	1.434	1.330–1.546	<0.001	1.185	1.092–1.285	<0.001
Peptic ulcer	1.114	0.998–1.244	0.053			
Hepatitis	0.775	0.662–0.906	0.001	0.959	0.813–1.130	0.616
Renal disease	2.359	2.153–2.585	<0.001	1.587	1.435–1.755	<0.001
Tumor location						
Colon	1			1		
Rectum	1.267	1.193–1.346	<0.001	1.491	1.395–1.594	<0.001
Tumor stage						
Stage I	1					
Stage II	1.704	1.539–1.887	<0.001			
Stage III	2.717	2.466–2.994	<0.001			
pT status						
T1	1			1		
T2	1.675	1.419–1.977	<0.001	1.3	1.089–1.551	0.004
T3	2.433	2.112–2.803	<0.001	1.524	1.300–1.785	<0.001
T4	3.977	3.446–4.591	<0.001	2.193	1.860–2.586	<0.001
pN status						
N0	1			1		
N1	1.504	1.399–1.617	<0.001	1.392	1.288–1.504	<0.001
N2	2.612	2.424–2.815	<0.001	2.381	2.192–2.587	<0.001
Histologic type						
Adenocarcinoma	1			1		
Mucinous	1.192	1.053–1.349	0.001	0.954	0.830–1.096	0.504
Signet ring cell	3.143	2.396–4.122	<0.001	1.839	1.360–2.486	<0.001
Histologic grade						
Well	1			1		
Moderate	1.291	1.178–1.414	<0.001	0.951	0.859–1.053	0.334
Poor	1.726	1.510–1.974	<0.001	1.055	0.906–1.229	0.488
Operative finding						
Obstruction	1.784	1.630–1.953	<0.001	1.258	1.136–1.393	<0.001
Perforation	1.811	1.567–2.093	<0.001	1.131	0.962–1.331	0.136
Preoperative values						
Hemoglobin						
<10	1			1		
≥10	0.617	0.575–0.663	<0.001	0.855	0.788–0.929	<0.001
Albumin						
<3.5	1			1		
≥3.5	0.406	0.376–0.439	<0.001	0.589	0.538–0.646	<0.001
CIPI						
<8	1			1		
≥8	2.262	2.121–2.411	<0.001	1.503	1.400–1.613	<0.001

CIPI = cancer-inflammation prognostic index.

## Data Availability

Due to privacy and ethical concerns, data details and how to request access are available from the corresponding author.

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
