# Peer review of "Preoperative Cancer Inflammation Prognostic Index as a Superior Predictor of Short- and Long-Term Outcomes in Patients with Stage I–III Colorectal Cancer after Curative Surgery"

_cancers, 2022, doi:10.3390/cancers14246232_

Round 1

Reviewer 1 Report

I dont have major comments to Authors

Author Response

Dear Reviewer,

The MDPI English editing service has reviewed our manuscript.

Regards,

Jeng-Fu You

Reviewer 2 Report

This study was an analysis of prognostic factors using a new marker called CIPI, which combines tumor factors and host factors. Although it is a new marker, it can be derived from CEA and NLR, so it is highly versatile in general clinical practice. Furthermore, the reliability of the data is high because the analysis is based on a very large number of cases.

However, there are some problems with acceptance.

1.         It's a basic question. Isn't it RFS that the author describes as DFS? An event of DFS includes all deaths, recurrences of colorectal cancer and development of other organ cancers. Survival rate with recurrence of colorectal cancer and all deaths as an event is RFS. That is Relapse free survival.

2.         I was surprised to find that the survival rate was stratified by CIPI difference at each stage. So here's the question. How do you think this CIPI should be applied clinically? Do you want to use it in your daily practice as a tool that surpasses TNM?

Author Response

Dear Reviewer,

The MDPI English editing service has revised our documents.

Our response to the reviewer's comments:

Response 1:

First, thank you for the precise terminology suggestion. We have changed "disease-free survival" (DFS) to "relapse-free survival" (RFS) in red font throughout the text.

Response 2:

Thank you for your insightful suggestions. This study demonstrates that the CIPI is a significant independent prognostic factor for resected CRC. The application of CIPI to various CRC predictive models requires additional research to ensure its utility; this will be the focus of our future work.

Reviewer 3 Report

This is a well done study regarding the Cancer Inflammation Prognostic Index  (CIPI) as a predictor for outcome in patients with stage I-III colorectal cancer after curative resection.  The study is well done and includes univariate and multivariate analyses.  The article is nicely organized and well written.  It should be a good contribution to the literature.

Author Response

(The authors gave the same response as above.)
